# *In vitro* proliferation of *Mytilus edulis* male germ cell progenitors

**Hajar Hosseini Khorami**, **Sophie Breton**, **Annie Angers** *

Département de Sciences Biologiques, Université de Montréal, Montréal, Québec, Canada

* annie.angers@umontreal.ca

## Abstract

Our understanding of basic cellular processes has mostly been provided by mammalian cell culture, and by some non-mammalian vertebrate and few invertebrate cell culture models. Developing reliable culture conditions for non-model organisms is essential to allow investigation of more unusual cellular processes. Here, we investigate how cells isolated from different tissues of the marine mussel *Mytilus edulis* thrive and survive *in vitro* in the hope of establishing a suitable laboratory model for the investigation of cellular mechanisms specific to these bivalve mollusks. We found that cells dissociated from mantle tissue attached to the culture vessels and proliferated well *in vitro*, whereas cells isolated from gills, although remaining viable, did not maintain divisions over three to four weeks in culture. We used antibodies against the germ-line marker DEAD-box helicase 4 (DDX4), also known as VASA, and the epithelial cell marker cytokeratin to distinguish different cell types in culture. DDX4-positive cells were predominant in 25-day-old cultures from male mantles. Cells from other tissues remained in low numbers and did not seem to change in composition over time. Overall, the culture conditions described here allow an efficient selection of male germ cells that could be used to study specific cellular mechanisms *in vitro*.

## Introduction

Biological studies of metazoans at the cellular level are mostly limited to vertebrates and a small number of invertebrate models. Yet, many fundamental cell mechanisms existed long before the divergence of Metazoa and may have evolved differently along divergent lineages. A good example of this is the complexity of apoptosis in mollusks that far exceeds the simple apoptotic network described in the model invertebrates *Caenorhabditis elegans* and *Drosophila melanogaster* [1]. Apoptosis in mollusks is, in fact, similar in complexity to apoptosis in vertebrates but also has unique features related to their response to environmental changes and pathogens [1].

In recent years, *in vitro*, cultivated molluscan cells have contributed significantly to many disciplines, including neurobiology, immunology, toxicology, functional genomics, and others. Most reports of successful attempts to culture molluscan cells indicated that primary cell cultures originating from various tissues, including hemolymph (hemocytes), heart, mantle, digestive gland and gill, can persist for extended periods *in vitro*, making them suitable systems

**Data Availability Statement:** Data for all figures and sample image files are publicly available from the figshare repository (https://doi.org/10.6084/m9.figshare.24805965).

**Funding:** AA RGPIN-2017-05054 National Sciences and Engineering Research Council

(NSERC) of Canada https://www.nserc-crsng.gc.
ca/ SB RGPIN-2019-04076 National Sciences and
Engineering Research Council (NSERC) of Canada
https://www.nserc-crsng.gc.ca/ The funders had
no role in study design, data collection and
analysis, decision to publish, or preparation of the
manuscript.

**Competing interests:** The authors have declared
that no competing interests exist.

for basic cell research (reviewed in [2]). Both short- and longer-term molluscan cell cultures
have been exploited for advancing the understanding of complex biological phenomena and
physiological systems [3]. One biological phenomenon specific to bivalve mollusks that could
greatly benefit from cell culture experiments to be better understood is their unique mitochon-
drial transmission system. Mitochondria are usually transmitted strictly maternally in animal
species, but many bivalve mollusks exhibit a very peculiar way of transmitting their mitochon-
dria, as they have been shown to keep the mitochondria of paternal origin in the male germ-
cell lineage [4–6]. This phenomenon has been dubbed Doubly Uniparental Inheritance (DUI)
and occurs in more than one hundred species distributed across twelve families of bivalve mol-
lusks [7]. The cellular basis allowing paternal mitochondria to be preserved specifically in the
germ-cell lineage of male individuals is still largely unknown. A cellular model would be a
valuable tool to advance our knowledge of the mechanisms underlying DUI and mitochon-
drial inheritance in general.

A good model system among bivalve mollusks is the genus *Mytilus*. *Mytilus* spp. are largely
distributed, widely studied as environmental sentinels, extensively cultured in commercial
mariculture (or mytiliculture) and they exhibit DUI. Cell cultures from different *Mytilus* spe-
cies have been used in both applied and basic research [3]. For example, hemocytes and gill
cells of *Mytilus galloprovincialis* have been used to assess cytotoxicity in several assays [8–11].
Longer-term cultures were established to look at the viability and cryopreservation of *M. edulis*
cells from different tissues [12–15]. Apoptosis, cell differentiation and adhesion were studied
in cultures of dissociated larvae cells from *M. trossulus* [16–19]. Of these, only Daugavet and
Blinova [14] used mantle dissociations, focussing on cells from the edge of the mantle there-
fore excluding gonads (gametogenesis takes place in this tissue in *Mytilus* spp.), and showed
that these cells remained viable for extended periods, but did not significantly proliferate.

Very few studies examined mantle cell and gonadal cell cultures in other bivalve species,
although earlier attempts demonstrated that the mantle of the sea clam *Paphia malabarica* was
a good source of viable cells in primary culture [20]. More recently, viable cell cultures were
established from gonads explants of the sea clam with DUI *Ruditapes philipinarum* [21]. These
cells proliferated for approximately 15 days before rapidly losing viability. Interestingly, some
of these cells expressed the germ-line marker VASA [21].

Here, we sought to determine suitable conditions to establish primary cell culture from
*Mytilus edulis*, selecting cells that attach and proliferate *in vitro*. Our goal was to establish a
model suitable for *in situ* and biochemical experiments. We focused on mantle (containing
gonad tissues) and gills dissected from male and female individuals to ensure a variety of cell
types. We found that *Mytilus edulis* cells survived better at higher temperatures with supple-
mentation of the culture media with calf serum and yeast extract. Cells extracted from the
mantle proliferated better than cells from the gills, and we obtained larger cell numbers from
male mantles. We determined that the major cell type present after 25 days in culture were
from the male germ-cell lineage. Cells obtained from gills or female mantle cells remained low
in number and did not seem to differentiate. To our knowledge, it is the first time that the sex
of the individuals from which the tissues are dissected is taken into account in such a study in
Mytilidae.

## Materials and methods

### Preparation of substrate

Since we wished to select adherent cells for further manipulations, we treated all culture dishes
and coverslips with synthetic amino acids to provide a suitable substrate for cell attachment.
Initial attempts using Poly-L-Lysine failed to facilitate cell attachment. We thus selected Poly-

D-Lysine (Gibco A3890401) and proceeded as follows. Borosilicate coverslips (Fisher Scientific 1254580) were sterilized by incubation in 95% ethanol for 1 hour. They were then washed with sterile ddH$_2$O and transferred into 24 well plates. When dried, coverslips were covered with 50 μg/ml Poly-D-Lysine (PDL) solution overnight, washed with sterile ddH$_2$O and allowed to air dry. All manipulations were carried out in a class II biosafety cabinet to ensure aseptic conditions.

## Sample collection and tissue preparation

Live *Mytilus edulis*, farmed by Prince Edward Aqua Farm, Canada, were purchased from the market no more than 24 hours before use. Individuals were cleaned of epibionts and washed with fresh water. They were wiped with 70% ethanol, briefly dried and opened with a sterile scalpel. Mantle tissue, including gonadal tissue, and gills of each mussel were dissected and transferred to dishes containing filtered artificial seawater ASW (11 mM CaCl$_2$, 10 mM KCl, 27 mM MgCl$_2$, 27 mM MgSO$_4$, 2 mM NaHCO$_3$, 200 mM NaCl plus 6 g/l dextrose) [22]. Mussel sex was identified by microscopic observation of gonadal smears for either eggs or sperm. Tissues were cut into small pieces of 1 to 2 mm$^2$. All fragments were rinsed 3 times with ASW supplemented with 100 U/mL penicillin, 100 μg/mL streptomycin, and 1.5 μg/mL Amphotericin B (Cytiva).

## Tissue dissociation and culture media

Basic culture conditions were optimized using mantle tissue dissociation from 10 male individuals. Tested media were composed of either ASW or Leibovitz 15 (L-15) dissolved in ASW, supplemented or not with bovine calf serum (BCS) alone or BCS and yeast extract (YE) (Table 1). We experimented with different dissociation techniques, namely gentle mechanical dissociation, incubation with 0.25% trypsin diluted in Instant Ocean Sea Salt (0.03 g/mL) and a combination of both. Mechanical dissociation almost systematically resulted in contaminated cultures and was rapidly abandoned. Thus, in most experiments, we used incubation with trypsin with gentle rotation for 15 minutes or until a cloudy suspension was formed. To stop trypsinization, L-15 containing BCS and antibiotics was added to the suspension. The suspension was then passed through two cell strainers with 100 μm and 40 μm mesh size, respectively, to form a more homogenous suspension and collected into a 50 mL falcon tube. Fragments on mesh were pressed with the bottom of a syringe gently and washed with ASW containing antibiotics to allow more cells to pass through the cell strainer. The suspension was then centrifuged for 7 minutes at 300 g at room temperature. The supernatant was discarded, and the cell pellet was washed three times with ASW plus antibiotics. Finally, cells were seeded at approximately 2×10$^4$ cells/well in Poly-D-Lysine-coated 24 well plates in triplicate in each culture medium. We experimented at 4 different temperatures (15, 18, 22 and 25˚C). As the results at 18˚C and 22˚C were generally comparable to those at 15 and 25˚C respectively, we

**Table 1. Media composition.**

|   | Base | Supplement |
|---|------|------------|
| 1 | ASW | – |
| 2 | ASW | 20% BCS |
| 3 | ASW | 20% BCS + 0.1% YE |
| 4 | L-15 + ASW | – |
| 5 | L-15 + ASW | 20% BCS |
| 6 | L-15 + ASW | 20% BCS + 0.1% YE |

used only 15 and 25 in the quantitative experiments. Media was completely renewed every 4–5 days to remove debris and unattached cells. To prevent contamination, penicillin (100 U/mL), streptomycin (100 μg/mL), and amphotericin B (1.5 μg/mL) were added to all media [23].

## Cell viability and proliferation

Cell number was assessed every five days by automatically counting stained nuclei. Nuclei were stained with 1.25 μg/mL Hoechst 33342 (Thermo scientific 62249) and counted from five randomly selected micrographs from each well every 5 days from day 1 to day 30 using 20X magnification on a fluorescent microscope (EVOS-M5000). Cell number for each well is the average number from the five micrographs, and each sample was seeded in triplicate. After imaging, cells were washed with fresh medium and returned to the incubator.

## Sex and tissue comparison

After selecting the best culture medium and dissociation method, we compared viability and proliferation of cells dissociated from male and female mantles and gills. Mantles and gills from 10 males and 10 females were obtained as above and cultured for 25 days at 25˚C in L-15 enriched with BCS and YE. Cells were counted as above.

## Immunofluorescence

To identify different cell types in culture, immunocytochemistry analyses were conducted on cells cultured from mantle and gill tissues of 3 males and 3 females after 2, 5, and 25 days in culture. Cells attached to poly-D-lysine-coated coverslips were washed with 1X phosphate buffered saline (PBS PH 7.4 ±0.1, 1.8 mM $KH_2PO_4$, 10 mM $Na_2HPO_4$, 137 mM NaCl, and 2.7 mM KCl), fixed with 4% paraformaldehyde for 30 minutes, rinsed 3 times with PBS, and permeabilized with 0.2% triton X-100 for 4 minutes. Cells were incubated for 30 minutes with 5% bovine serum albumin (BSA) mixed with 5% normal goat serum (NGS) as the blocking solution, washed three times with PBS, and incubated overnight in a humid chamber with 3 μg/ml anti-DDX4/MVH antibody (Abcam ab13840) germ-cell progenitor marker, and 10 ug/ml anti-Cytokeratin Pan Type I Monoclonal Antibody (AE1) (Invitrogen MA5-13144), epithelial cell marker. Cells were then washed with PBS and incubated with secondary antibodies Goat anti-Mouse IgG (H+L) Alexa Fluor™ 594 and Goat anti-Rabbit IgG (H+L) Alexa Fluor™ 488 (Invitrogen) for one hour at room temperature. Finally, slides were washed with PBS, mounted with Fluoroshield Mounting Medium with DAPI (Abcam ab104139), and imaged with an epifluorescent microscope (EVOS M5000) or confocal microscope (Zeiss LSM 800). Staining specificity was ascertained by using both antibodies separately and omitting the secondary antibodies (not shown).

## Statistical analysis

To examine cell proliferation in different conditions (temperature, media and tissue of origin), we performed general linear models (GLMs) using cells count data as response variables, with a negative binomial distribution and logarithm link function. To ascertain proliferation in specific conditions, we performed ANOVA followed by Tukey's multiple comparison tests (S1 and S2 Figs). Statistical analyses were conducted using the statistical environment R version 4.3.1 [24] with the MASS package [25] for the GLMs.

## Results

### Survival and proliferation of *M. edulis* cells in culture

To determine the best dissociation method and culture medium, we used mantle tissue of male mussels, which is abundant and heterogeneous as it contains the gonads in the form of many ducts where gametogenesis occurs. We first tried mechanical dissociation, which seemed to provide higher cell number, but rapidly abandoned this approach since most cultures became contaminated. Using trypsin to dissociate the cells sporadically yielded contaminated cultures, but in most cases, cultures could be kept free of contamination for several weeks. This is possibly attributable to the enzyme itself, as it has been shown to disrupt biofilms and improve antimicrobial agent efficiency [26]. In the end, a combination of both mechanical and enzymatic treatments was preferred.

To assess the best supplementation and culture conditions, six different culture media (Table 1) were tested at different temperatures (15, 18, 20 and 25˚C). In all media tested, we observed a steady decline of cell populations when the cultures were kept at 15˚C (Fig 1B). After empirical testing of the different temperatures, it appeared that mantle cells proliferated best at higher temperatures (Fig 1A and 1C). At 25˚C, the presence of nutrients provided by L-15 was required to maintain cell viability, as all media based on artificial sea water alone could not support cell populations. When the media was supplemented with BCS, the number of attached cells increased with time, showing that cells were dividing ($p < 0.001$; Figs 1C and S1). Note that there was no significant increase between Day 1 and Day 5 in any of the tested media, likely attributable to the loss of most cells that failed to attach after the first change of culture medium. The addition of both BCS and YE seemed preferable as considerably less variability was obtained in this medium. In these conditions, cell numbers doubled in 4 to 5 days and continued to increase until Day 20 in culture (Figs 1A and 1C and S1). At this point, cells had reached confluency, and only marginal growth was observed.

As an alternative source of growth factor supplements, mussel serum was prepared from *M. edulis* mantle tissue following the method described by Vandepas et al. [23]. However, this supplementation was not successful as even a low quantity of mussel serum added to the media resulted in the formation of a high viscosity film on the culture dish, preventing cell attachment and growth.

### Proliferation of cells from different tissues

Having selected L-15 enriched with BCS and YE at 25 ˚C, we compared the proliferation of cells dissociated from male and female mantles and gills. Ten samples were obtained for each tissue and sex. Of these, only cells dissociated from male mantles could attach to the surface and establish cultures containing an appreciable number of cells, but viable cell cultures were obtained from all tissues, albeit in low abundance. Both male and female gill cells provided similar results, with a low number of cells attaching to the dish and proliferating poorly (Figs 2 and S2). The cell population for gill cells approximately doubled during the course of the experiment. The proliferation was more variable for female mantle cells, with some cultures thriving, but others showing very low proliferation rates. On average, these cells doubled in approximately 6 to 7 days (from interpolation) and plateaued after 15 days without reaching confluency. The total population increased approximately 3 folds during the course of the experiment. Fig 2 shows that the number of cells obtained from male mantles was several orders of magnitudes higher than for the other tissues. The relative growth of attached cells for male mantle cells was almost 4 folds over the course of the experiment, and showed significant increase until Day 25 (Figs 2 and S2).

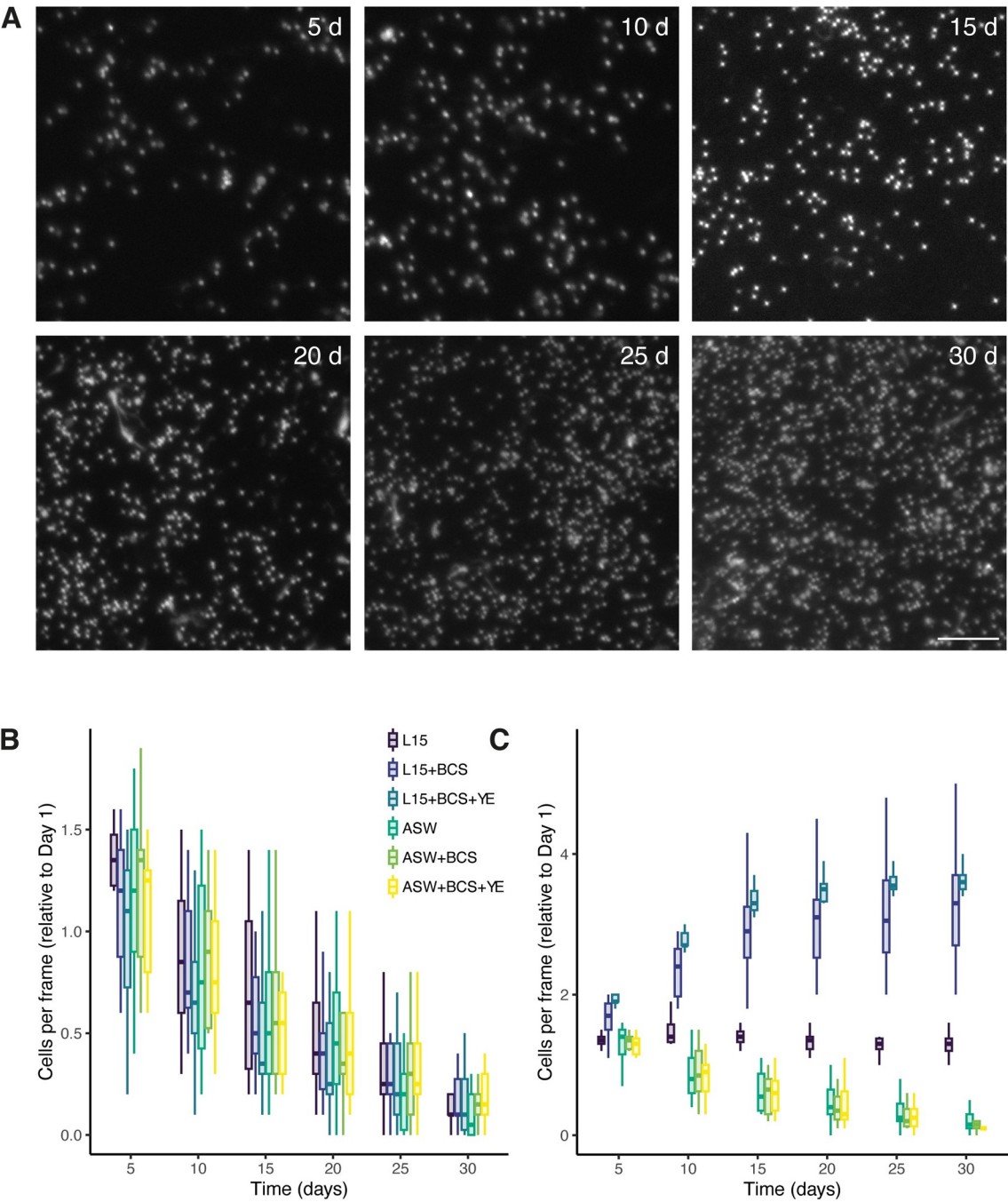

**Fig 1. Male mantle cell viability and proliferation in different culture media.** A) Examples of images obtained for counting cells from one culture at the indicated time point. Nuclei were stained with Hoescht dye and cells were counted using automatic detection. Five frames from triplicate cultures were counted for every sample. For space concerns, only a fraction of the complete image is shown. Scale bar = 25μm. B, C) Relative number of cells per frame compared to the number of cells that remained attached after the first change of media (Day 1) in each medium at 15 (B) and 25°C (C) after the indicated number of days in culture. Within each box, bold horizontal lines denote median values; boxes extend from the 25th to the 75th percentile of each group's distribution of values; vertical extending lines denote the 5th and 95th percentiles.

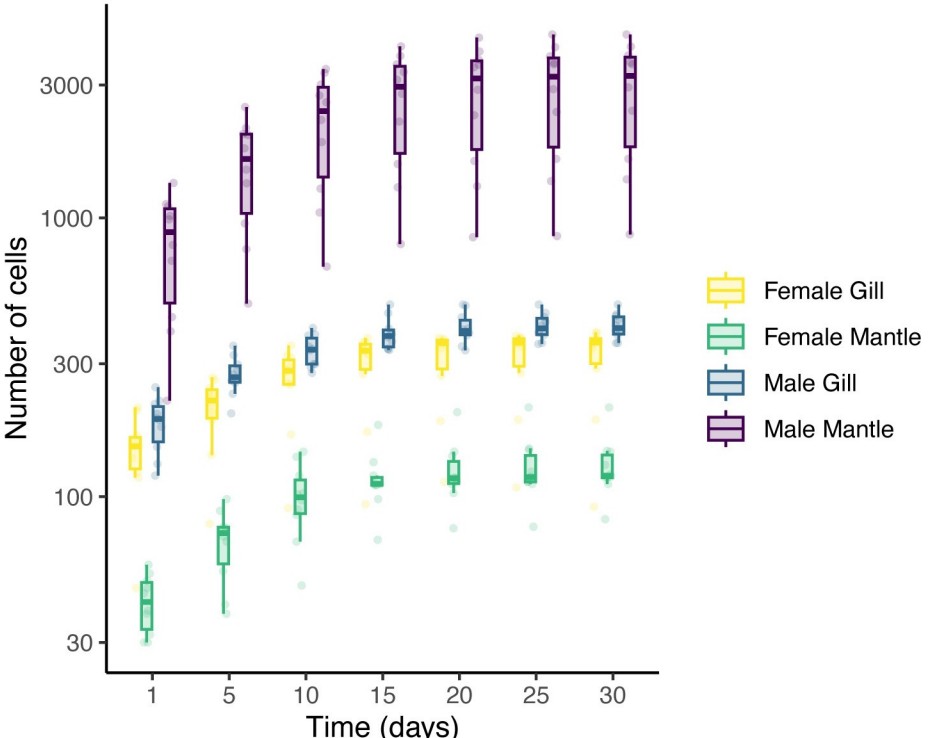

**Fig 2. Average number of cells per microscopic frame (0.3 mm²) from dissociated male and female mantles and gills after the indicated number of days in culture.** 5 microscope fields were counted from three replicates from 10 males and 10 females. L-15 medium dissolved in artificial sea water supplemented with antibiotics and enriched with 20% BCS and 0.1% YE was used as a culture medium, changed every four days. Within each box, bold horizontal lines denote median values; boxes extend from the 25th to the 75th percentile of each group's distribution of values; vertical extending lines denote the 5th and 95th percentiles. The number of cells for each individual also appears as small lighter dots.

## Cell morphology and markers expression

We used immunofluorescence to detect cells expressing DDX4 (VASA) and cytokeratin to begin identifying the cell types selected in these culture conditions. DDX4 is an RNA-dependent helicase expressed in animal germ cells and has been widely used as a marker of germ-cell progenitors in a wide variety of species [27]. We used a pan-cytokeratin antibody as a marker of epithelial cells. Invertebrate intermediate filament protein expression is not as complex as vertebrates, but epithelial cells have been shown to express proteins sharing significant homology with the conserved regions of vertebrates' type A and B cytokeratin and to react with antibodies against these proteins [28].

We labelled cells cultured from male and female mantles and gills after different times in culture. To assess cell morphology, phase contrast images were also captured. Most cells present in male mantle cultures were small, rounded cells of approximately 2.5 μm diameter (Fig 3A–3D). Larger round cells ranging from 4 to 6 μm in diameter were also frequently observed. More rarely, elongated fibroblast-like cells attached to the coverslip.

In cultures obtained from female mantle samples, small cells were also present, although they were more ovoid and slightly larger (4 to 5 μm) than those present in male mantles (Fig 4A and 4B). Elongated cells like those observed in male samples could also be found in cell cultures of female mantles (Fig 4C).

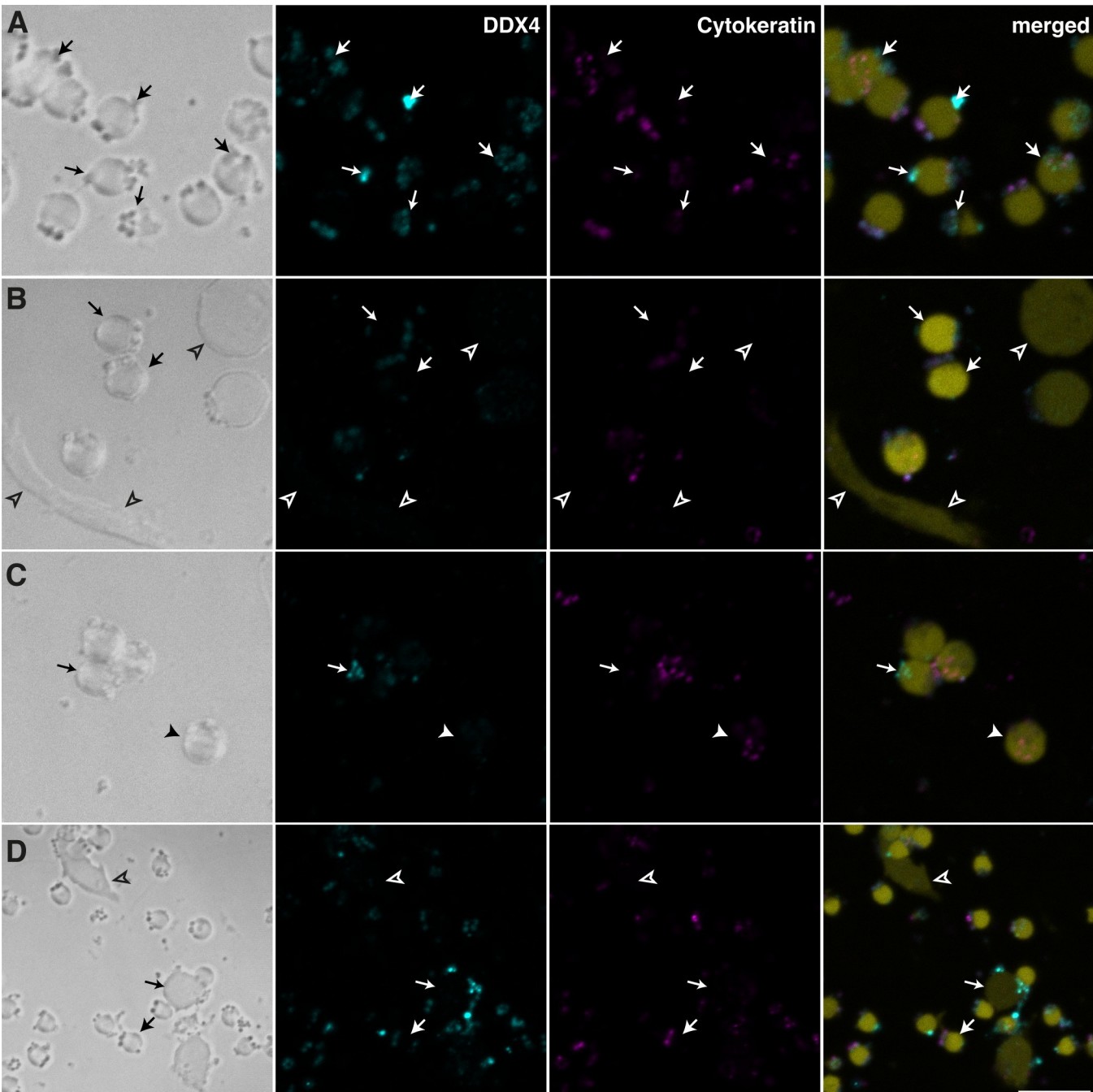

**Fig 3. Different cell morphologies and expression of DDX4 and cytokeratin in cells from male mantle after 5 days in culture.** (A-D) Phase contrast and immunofluorescence staining for DDX4 (cyan) and cytokeratin (magenta) showing cells positive for both antigens (type A cells, large arrows), DDX4 only (type B cells, small arrows), cytokeratin only (type C, filled arrowheads), or unlabelled cells visible by weak autofluorescence (type D, empty arrowheads). Nuclei were counterstained with DAPI (coloured in yellow). (A) High magnification of typical type A and type B cells of approximately 2.5 μm in diameter and positive for both antibodies or DDX4 only. (B) Similar rounded cells weakly positive for both antibodies and DDX4 only, a larger cell (5.5 μm) and two elongated cells (~7.5 μm in length) negative for both antibodies. (C) Small rounded cells of each type. (D) Lower magnification showing small, rounded cells and one bigger, oval cell (~5 μm) positive for both antibodies, one large (~5 μm) cell positive for DDX4 only, and one unlabelled elongated cell (11 μm). Scale bar represents 5 μm in A,B and C and 10 μm in D.

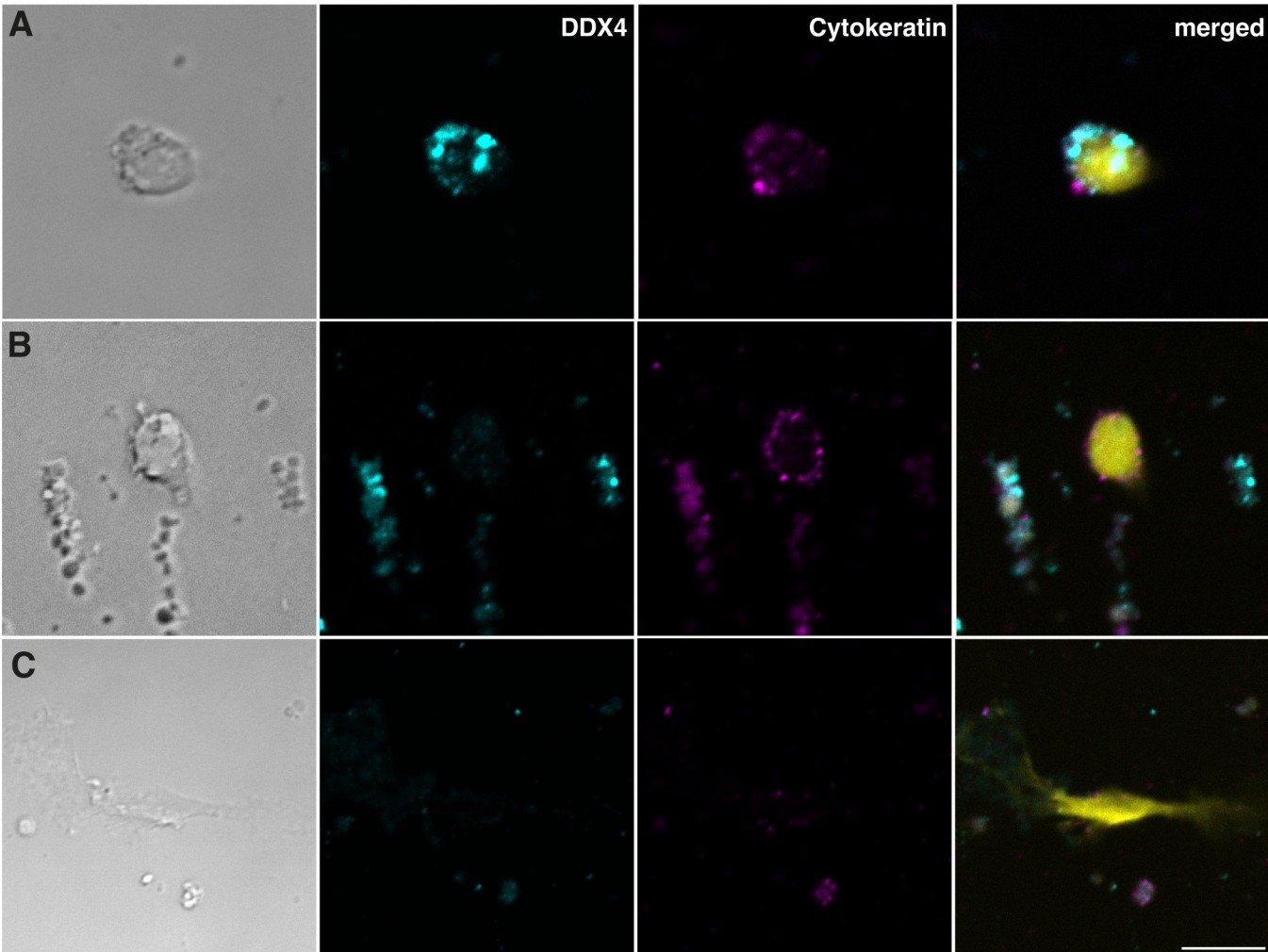

**Fig 4. Different cell morphologies and expression of DDX4 and cytokeratin in cells cultured from female mantle after 5 days in culture.** (A-C) Phase contrast and immunofluorescence staining for DDX4 (cyan) and cytokeratin (magenta), with nuclei counterstained with DAPI (yellow). (A) Small ovoid cell positive for DDX4 and cytokeratin. (B) Small ovoid cell positive for cytokeratin only. Note the debris surrounding the cell that was often present in cultures from female samples. (C) Large, elongated cells negative for both antibodies. Scale bar represents 5 μm in A and B, and 7.5 μm in C.

Cells obtained from male and female gills were similar to those from the female mantle. More than half of the cells from male and female gills were large, round cells with 4 to 7 μm diameter. The second most common gill cells were small, round cells up to 3 μm in diameter. The presence of ovoid and elongated cells in male and female gill culture was rare and ranged from 4 to 7.5 μm in diameter.

Cells were identified as type A when positive for both DDX4 and cytokeratin, type B if positive only for DDX4, type C if positive only for cytokeratin or type D when neither marker was present. Tentatively, type A would represent germ cell precursors in early differentiation stages [29], type B more differentiated germ cells [21], type C would represent epithelial cells and type D any other cell type that could be present in the culture.

Most of the small, rounded cells present in male mantle cultures tended to be positive for both DDX4 and pan-cytokeratin in young cultures, although some cells were positive for only one or the other. Larger round cells were also often positive for both markers at early stages. Some elongated cells could be seen by autofluorescence and transillumination but expressed

neither DDX4 nor cytokeratin (Fig 3). Fig 4 shows that cell obtained from female mantles positive for DDX4 and cytokeratin or cytokeratin only tended to be larger and more ovoid than male mantle cells (Fig 4A and 4B). The presence of unlabelled elongated cells was also noted in these cultures (Fig 4C).

Cells were monitored after 2, 5 and 25 days in culture. Using the type A to D nomenclature, we compared the evolution of cell populations with time in culture. The number of type A cells remained more or less the same in every cultures, whereas types B, C and D increased in cultures from male and female gills and from female mantle. Male mantle cells differed, showing a large increase in the number of type B cells, while all other cell types declined over time (Fig 5). The proportion of each cell type changed with time and the tissue of origin. Type A cells tended to be the most abundant cells attaching to the coverslips after two days in culture, representing between 35 and 75% of all cells present (Fig 5). In male gills, type A cells were slightly more abundant (35%) than type D, with no staining (30%). After 5 days, type A cells remained the most abundant in cultures of male and female mantle cells, representing 70% and 63% of cells, respectively (Fig 5A and 5B). In gill cells from both sexes, types C and D were more abundant on Day 5, representing 25 and 48% of cells in samples from males and 40 and 26% of cells in samples from females (Fig 5C and 5D). After 25 days in culture, the proportions of type B cells expressing only DDX4 and type A cells expressing both markers had shifted in cultures originating from male mantle cells, with 26% type A and 66% type B (Fig 5A). This shift did not occur in cultures originating from female mantle cells nor from gill cells of both sexes, the proportions of each cell type remaining more or less the same (Fig 5B–5D).

We also compared the morphology of the most abundant cells expressing only DDX4 in older male mantle cultures (i.e. type B more differentiated germ cells) to mature sperm cells. Specifically, we stained cells dissociated from male mantle that had been cultured for 44 days and spermatozoa collected from mature males with DDX4 and ATP5, a mitochondrial marker (Fig 6). Mature spermatozoa in *Mytilus* harbour a characteristic group of five mitochondria arranged in a circle at the base of the flagella. These can be seen in the micrograph of Fig 6B and 6C, along with the flagella and acrosome of mature sperm cells. Whereas DDX4 positive cells in mantle cultures were of similar diameter and showed the same peculiar arrangement of mitochondria on one pole (Fig 6A), they are not fully differentiated sperm cells as they do not have a flagellum nor acrosome. However, it is reasonable to suppose that they have differentiated into spermatids (see Discussion).

## Discussion

### Culture conditions and cell proliferation

We tested different cell dissociation methods, as well as different media compositions at different temperatures using *M. edulis* male mantle samples to select the best conditions for primary cultures of adherent mussel cells. The mantle is a sheet of dorsal epidermis extending along both shells in bivalve mollusks. It is thought to fulfill several functions, including the secretion of calcium carbonate to generate the shell and sensory functions. It also contains muscle cells [30]. In *Mytilus*, the mantle comprises connective and gonadal tissues [31]. It thus contains many different cell types, of which some were shown to proliferate *in vitro* [16]. For example, mantle cells were maintained viable for up to 22 months in L-15 dissolved in artificial seawater supplemented with 2% bovine serum and kept in sealed plates at 10°C without changing the medium. These cells attached to the bottom of the dish after approximately 6 days and remained viable but did not significantly proliferate [14]. The present study found that mantle cells obtained using a combination of both mechanical and enzymatic dissociation rapidly attached to the matrix. The addition of BCS was essential to sustain proliferation, and yeast

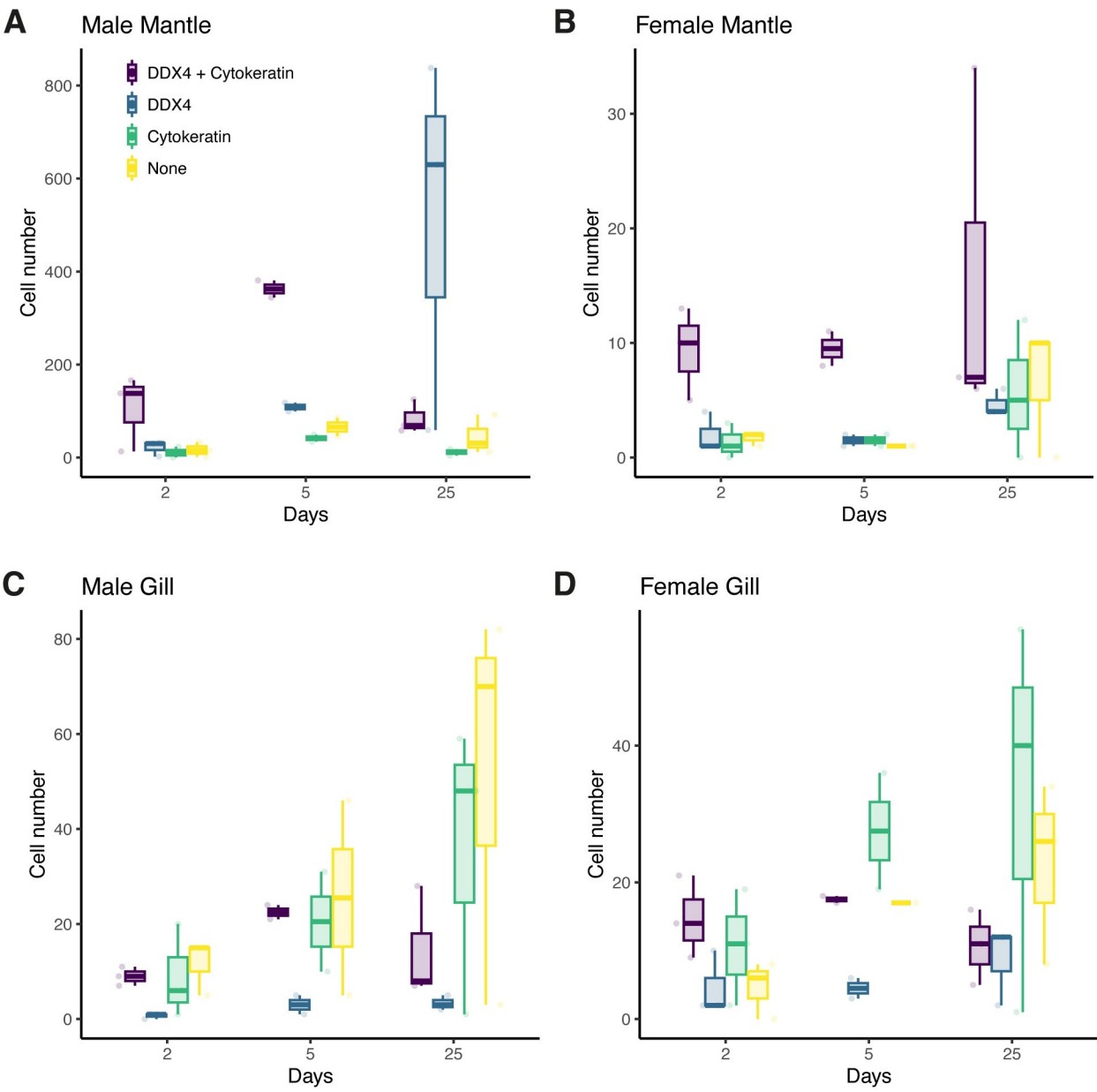

**Fig 5. Change in marker expression over time in cultures originating from mantle and gill cells.** The number of each cell type over time as determined by detection of DDX4 and cytokeratin immunofluorescence in cultures of (A) male mantle cells, (B) female mantle cells, (C) male gill cells and (D) female gill cells. The number of unstained cells was obtained by counting the DAPI stained nuclei. Cells were counted from at least 10 captured frames at 100X magnification in 3 independent experiments (2 for male samples on Day 5).

extract provided stability to the culture media, reducing the variability between cultures. Yeast extract was previously used in culture media as a source of vitamins and other supplements [32]. It is thus reasonable to presume that its addition complements the L-15 formulation and provides unknown useful components for mussel cell maintenance [33].

It was somewhat surprising that the proliferation of mantle cells in culture could only be obtained at relatively high temperatures (room temperature up to 25˚C). All cultures kept at 15˚C, no matter the medium used, showed a clear reduction in cell numbers with each media change, suggesting that the cells were losing viability or could not maintain their adhesion to

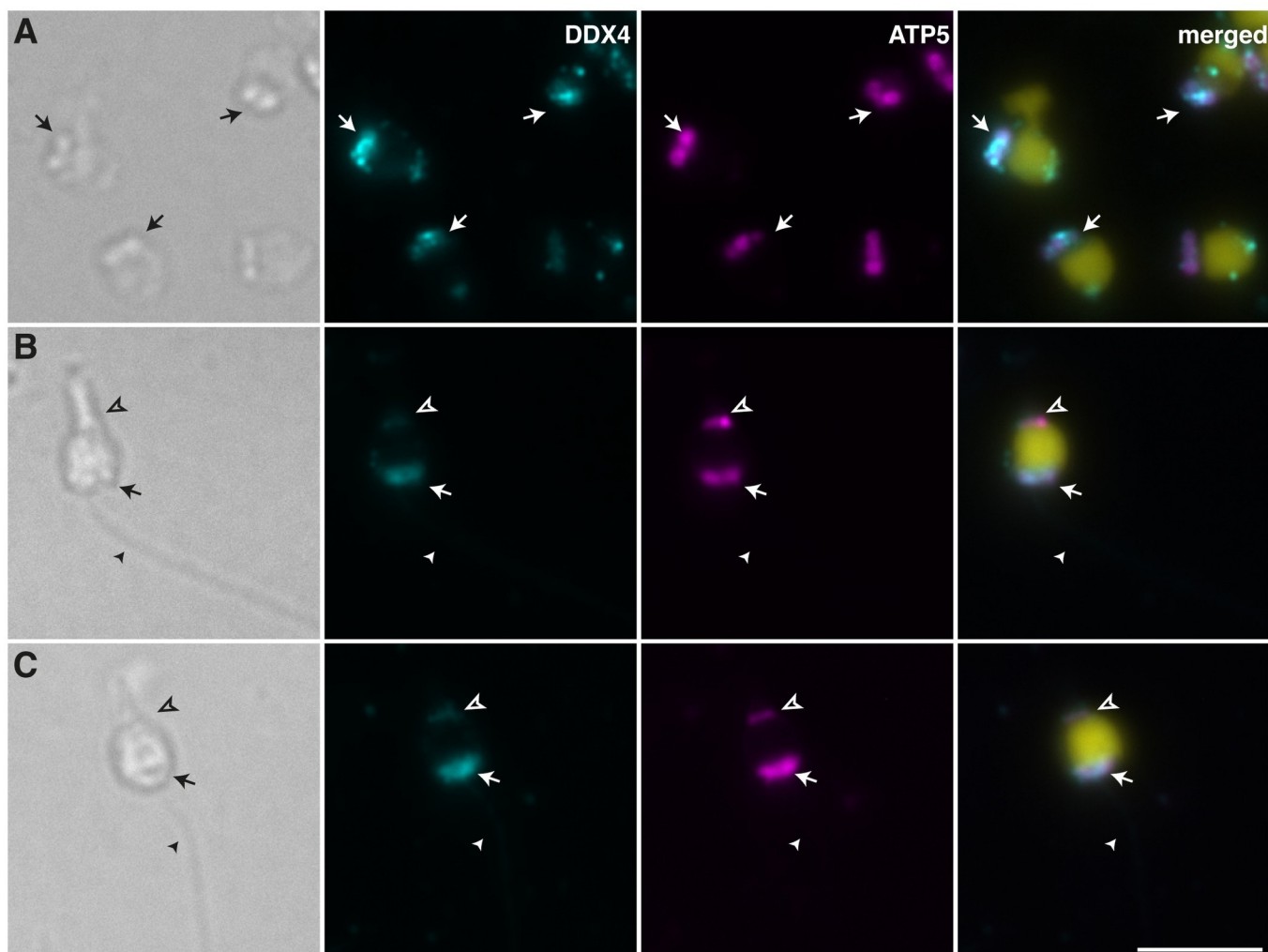

**Fig 6. Mantle cells and spermatozoa stained with DDX4 and ATP5.** Light micrographs and immunofluorescence staining for DDX4 (cyan) and ATP5 (magenta), with nuclei counterstained with DAPI (yellow). (A) Cells dissociated from male mantle and cultured for 44 days. The arrow points at ATP5 stained mitochondria assembled at the base of the cells. (B and C) Example of mature spermatozoa similarly stained showing ATP5-positive mitochondria (arrow) at the base of the flagellum (arrowhead). The elongated acrosome (empty arrowhead) is also labelled by ATP5 and DDX4. Scale bar represents 5 μm.

the plate. In most previous studies, cell cultures were kept at temperatures ranging from 10 to 18˚C [10, 14, 15, 18, 21, 34, 35]. In our case, mantle cells remained viable for several days at 25˚C and significantly proliferated when given adequate culture conditions. Although all our experiments were finally carried out at 25˚C, we later found that keeping the cells at room temperature (~22˚C) also provided adequate culture conditions and somewhat limited contamination. Interestingly, these cells can be submitted to immunofluorescence, as we have shown, and will provide an interesting model to study in more detail molluscan cell physiology and differentiation.

After having established the best dissociation method and culture conditions with male mantle cells, we examined female mantle cells and male and female gill cells. The number of cells obtained was clearly higher in samples prepared from the male mantles than from the gills. There was also a clear difference between samples prepared from males and females, with the culture of male mantle cells yielding, on average, 60-fold the number of cells from any other tissue tested. A possible explanation for this discrepancy between male and female

mantles is the presence of mature and immature oocytes that contain large amounts of lipids and proteins and that seem to rapidly disrupt in the media, polluting the dish and preventing cell attachment. This potential problem was avoided in [14] by using only the edge of the mantle since the gonads are located at the basis of the tissue in *M. edulis*. Attaallah [21] did not report any problem either for mantle cells from the marine clam *R. philippinarum*, but in this species, the mantle and the gonad are distinct and can be dissected separately. Nevertheless, they reported that oocytes may be kept in culture for this species, which does not seem to be the case here. The key difference in their experiments is that non-adherent cells were also considered [21]. Interestingly, although scarce in number, female mantle cells proliferated at a rate almost comparable to male mantle cells. Since, as discussed below, cells proliferating in the male mantle samples are likely of the germ cell lineage, it will be interesting to determine if conditions allowing the selection of more viable cells from female mantle can be established.

## Cell type and differentiation

Morphological examination of cultures originating from male and female mantle and gill tissues showed that most cells remained rounded, even after attachment to the substrate. Occasionally, different morphologies were detected. After two days in culture, about 80% of the attached cells expressed either DDX4, cytokeratin, or both, possibly indicating the presence of pluripotent cells [21, 29]. Interestingly, the cultivation of *M. trossulus* larval cells (trochophore larvae ~24 hpf) allowed to assert that mussel embryonic/pluripotent cells can differentiate into a definite cell type under specific conditions *in vitro*, namely muscle and neuron-like cells in that case [18]. Although *M. trossulus* is genetically distinct from *M. edulis*, they remain very similar morphologically and ecologically [36]. Otherwise, dissociation of gill cells from *M. galloprovincialis*, also closely related to *M. edulis*, produced approximately 58% ciliated epithelial cells, the rest being non-ciliated epithelial cells and hemocytes [37]. This ratio is consistent with what was obtained here, where about 50% of gill cells expressed cytokeratin alone or combined with DDX4 in early cultures. The proportion of cytokeratin-expressing cells and cells not expressing any of the selected markers increased with time.

The situation was different in cells extracted from mantle samples. In female samples, the proportion of cells expressing both DDX4 and cytokeratin remained well above 50% at all times points, suggesting that putative pluripotent cells remained abundant in these cultures. The male mantle samples diverged from the others as cells expressing only DDX4 became largely dominant after a few weeks in culture. These results are similar to what was reported in cells extracted from explants of *R. philippinarum* gonads [21], strongly pointing to the gonadal characteristics of the cells proliferating in male mantle cells. The culture system we put forward here thus clearly supports *M. edulis* germ cells' selection, proliferation, and possibly differentiation. These findings provide an exciting possibility and a useful system to study early spermatogenesis, especially in a DUI species where male and female gonads carry different mitochondrial genomes.

The arrangement of mitochondria on one pole of the cells and the mitochondrial localization of DDX4 suggests that cells proliferating in male mantle cultures could be at an intermediate/late stage of sperm differentiation. Indeed, in the earliest stages of spermatogenesis until early spermatids, the mitochondria in *Mytilus* spp. are rather uniformly distributed throughout the cytoplasm. It is only from the mid-spermatid to the late-spermatid stage that mitochondria are reduced to five in number, and restricted in position to one juxtanuclear region which will eventually become the sperm middle piece [38]. Moreover, previous studies showed that the mitochondria of maturing sperm cells in bivalve species uptake DDX4 (VASA) [39].

That said, small, rounded cells positive for both DDX4 and pan-cytokeratin in young male mantle cultures also seemed to present this arrangement of mitochondria on one pole of the cells (Fig 3), suggesting that *Mytilus* male germ cells may express keratin until the spermatid stage or during the spermatid stage. For example, certain forms of keratin are expressed during mammalian spermatogenesis, with one group present in spermatogonia, spermatocytes, and spermatids. In contrast, another form is expressed only during the elongation and condensation of the spermatid nucleus [40]. With this in mind, one could speculate that most of the small, rounded cells positive for both DDX4 and pan-cytokeratin in young *Mytilus* male mantle cultures would be mid-spermatids while cells expressing only DDX4 that became largely dominant after a few weeks would be late spermatids. It would be interesting to determine if they could further differentiate into spermatozoa, given adequate conditions.

Indeed, male germ cells in both vertebrates and invertebrates can differentiate *in vitro* to produce motile sperm under certain conditions [41–44]. *In vitro* spermatogenesis has been achieved in several types of fish, and it was found that the addition of 11-ketotestosterone to the culture medium was essential to stimulate spermatogonial stem cell mitosis and complete spermatogenesis [45]. Other studies showed that the cocultivation of isolated germ cells with feeder cells mimicking Sertoli cells also supported successful spermatogenesis [46–48]. In mice, fetal bovine serum was indispensable to induce the differentiation of spermatogonia to haploid round spermatids but also suppressed the formation of elongating spermatids or spermatozoa. Using neonatal mouse testes explants and serum-free culture media, spermatids and sperm were obtained that resulted in healthy and reproductively competent offspring through microinsemination [49]. Presumably, germ cells obtained from *M. edulis* mantle could be brought to full sperm maturation by modifying the media supplementation to support spermatogenesis better.

## Conclusion

We have tested a number of conditions to establish the primary culture of cells extracted from different tissues in *Mytilus edulis*. Cells from female and male mantles and gills showed a moderate level of proliferation and remained viable for several weeks. Different cell types were present, and *in vitro* conditions seemed to favour epidermal and unknown cell types based on cytokeratin expression, except in preparations obtained from male mantles. Cells extracted from male mantles provided the best proliferation rate. Based on DDX4 expression, they differentiated into germ cells and proliferated well *in vitro*. This culture system can thus provide viable and proliferative cells that may be used to explore spermatogenesis in bivalve mollusks and fundamental questions regarding the unique system of doubly uniparental inheritance of mitochondria in bivalves. The described protocol could be applied in the future to characterize various cell types and organelles of DUI species' somatic and gonadal tissues. Applying the combination of these methods and genetic testing techniques on established cells may help clarify the level, tendency, and reason for heteroplasmy in DUI species.

## Supporting information

**S1 Fig. Statistical analysis of cell number in different media.** A) GLM model with a confidence interval for cell number in different media. There is no overlap of the confidence intervals at 25˚C, showing that cell growth is positive and significantly different in L15_BCS and L15+BCS+YE compared to L15 alone. Growth is negative in all other tested conditions. B) Tukey's comparison of mean cell number at different times for all media tested at 25˚C. Significance is shown only for consecutive measures. ns non significant, $^*p < 0.05$, $^{**}p < 0.01$, $^{***}p<0.001$.
(PDF)

**S2 Fig. Statistical analysis of cell number from different tissues.** A) GLM model with a confidence interval for cell number and growth obtained from different tissues. Cells from male and female mantle increased more over time than cells from male and female gills. There is no overlap in the models, meaning that all cultures behaved differently. B) Tukey's comparison of mean cell number at different times for all tissues tested. Significance is shown only for consecutive measures. ns non significant, $^*p < 0.05$, $^{**}p < 0.01$, $^{***}p < 0.001$.
(PDF)

## Acknowledgments

We thank the anonymous reviewers for their helpful comments that contributed to the improvement of the manuscript.

## Author Contributions

**Conceptualization:** Sophie Breton, Annie Angers.

**Data curation:** Annie Angers.

**Formal analysis:** Hajar Hosseini Khorami, Annie Angers.

**Funding acquisition:** Sophie Breton, Annie Angers.

**Investigation:** Hajar Hosseini Khorami, Annie Angers.

**Methodology:** Hajar Hosseini Khorami, Sophie Breton, Annie Angers.

**Project administration:** Annie Angers.

**Resources:** Sophie Breton, Annie Angers.

**Supervision:** Sophie Breton, Annie Angers.

**Validation:** Sophie Breton, Annie Angers.

**Visualization:** Sophie Breton, Annie Angers.

**Writing – original draft:** Hajar Hosseini Khorami.

**Writing – review & editing:** Hajar Hosseini Khorami, Sophie Breton, Annie Angers.

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
