## [Decision Letter · Decision Letter 0]

25 Oct 2023

PONE-D-23-29958*In vitro* proliferation of *Mytilus edulis* male germ cell progenitorsPLOS ONE

Dear Dr. Angers,

Thank you for submitting your manuscript to PLOS ONE. After careful consideration, we feel that it has merit but does not fully meet PLOS ONE’s publication criteria as it currently stands. Therefore, we invite you to submit a revised version of the manuscript that addresses the points raised during the review process.

We look forward to receiving your revised manuscript.

Kind regards,

Gao-Feng Qiu

Academic Editor

PLOS ONE

Journal Requirements:

2. Thank you for stating the following in the Acknowledgments Section of your manuscript: "This work was supported by the Natural Sciences and Engineering Research Council (NSERC) (grant no RGPIN-2017-05054 A.A. and RGPIN-2019-04076 S.B.)."

Please remove any funding-related text from the manuscript and let us know how you would like to update your Funding Statement. Currently, your Funding Statement reads as follows: "AA RGPIN-2017-05054 National Sciences and Engineering Research Council (NSERC) of Canada https://www.nserc-crsng.gc.ca/

SB RGPIN-2019-04076 National Sciences and Engineering Research Council (NSERC) of Canada https://www.nserc-crsng.gc.ca/

Reviewers' comments:

Reviewer's Responses to Questions

**Comments to the Author**

1. Is the manuscript technically sound, and do the data support the conclusions?

Reviewer #1: Yes

Reviewer #2: Partly

2. Has the statistical analysis been performed appropriately and rigorously? 

Reviewer #1: Yes

Reviewer #2: Yes

3. Have the authors made all data underlying the findings in their manuscript fully available?

Reviewer #1: Yes

Reviewer #2: Yes

4. Is the manuscript presented in an intelligible fashion and written in standard English?

Reviewer #1: Yes

Reviewer #2: Yes

5. Review Comments to the Author

Reviewer #1: The submitted manuscript is appropriately constructed and meets the criteria of Plos One editorial board. I believe that it would be of interest to the readership of Plos One, and would be suitable for publication after some minor revisions.

Introduction: The last paragraph in the introduction should give the reader more insight into your study. Based on previous studies, what are your hypotheses/expectations for each thing you examine?

Discussion: The authors need to improve the discussion section, focusing on this study's importance and relevancy.

Conclusion: Add suggestions and the actual necessities for future study.

Reviewer #2: Title: In vitro proliferation of Mytilus edulis male germ cell progenitors

Proliferation: Only markers have been used to identify germ cell progenitors. Whether any sub-cultures could be done is not mentioned. No experiments to measure actual proliferation (like BrdU assay) is done. BrdU staining or assay should be done during different days of incubation to show that cells are actually proliferating in vitro. Mere presence of germ line markers cannot verify if actual proliferation is happening or not. So, according to me, manuscript does not do justice to the title.

Inconsistency: Inconsistency in methodology and results- Only 2 temperature mentioned in methodology, but results talk about 4 temperatures tested, but then goes on to elaborate only about two specific temperatures 15 and 25 oC. Methodology does not describe the three cell dissociation methods tested, but only the final chosen method is described.

Significant difference obtained or not?

(a) In the figure 1B and C, significant differences between days is not mentioned. A graph with mean and standard deviation, and significance values denoted as letters will be easier to interpret. The current graphs do not show any symbols to indicate if there were significant differences, and the graphs show overlapping values between the different days of observation (see figure 1C), creating doubt if there was actually any significant difference (i.e. if there was actually any significant cell proliferation). The authors say that cell proliferation (male mantle) occurred only until Day 15 like has been previously reported by Attaallah et al. 2020. In that case, what new findings do authors have to contribute?

(b) Fig 2 A and B: In the study based on cell proliferation rate in different sex, the results say that female mantle cells also proliferated till Day 15. So, what is the difference in cell proliferation rate between male and female mantle cells? On what terms do you say that male mantle cells are better? Also, significant differences are not plotted on the graph. Also, many values in female mantle cells overlap with that of male mantle cells. The graph will be clearer with mean and standard deviation plots. You can keep either figure 2A or figure 2B. Two graphs convey more or less the same matter.

(c) Immunofluorescence: Fluorescence measured on 1, 5 and 25. It is said that proportion of germ cells increased on Day 25, whereas cell counting experiment says cell proliferation occurred only until Day 15. These may be confounding results. It may not be termed proliferation, but just a shift in cell population. So, this statement is inappropriate “DDX4-positive cells were the only cells that significantly proliferated in culture, and only from male tissue samples”. Keep either Figure 5 or 6. Two graphs convey more or less the same matter. Again, letters to denote significant differences among groups are required.

Antibiotics: Wide literature including personal experience shows that Amphotericin B is toxic to cells in vitro. How could you manage a long-term cell culture with Amphotericin B?

Figure 3 is difficult to interpret using the arrowhead legends. Example, Type A cells are supposed to exhibit both antigens, but I cannot see both antigens in the representative picture they have shown in Figure 3 (Panel A). May be both antigens are present on the cells, but the picture is not clear. Similarly for other images.

Figure 7 is a beautiful depiction of cell differentiation in vitro

Overall comment: the effort is worth appreciating but the results seem to be confounding, and the title and conclusions are misleading.

6. PLOS authors have the option to publish the peer review history of their article (what does this mean?). If published, this will include your full peer review and any attached files.

Reviewer #1: No

Reviewer #2: No

---

## [Author Response · Author response to Decision Letter 0]

16 Dec 2023

Dear Editor,

Thank you for allowing us to submit a revised and improved manuscript. We sincerely appreciate your feedback and thank the reviewers for their time and suggestions. We believe we have addressed most comments as detailed below.

Reviewer 1

The submitted manuscript is appropriately constructed and meets the criteria of Plos One editorial board. I believe that it would be of interest to the readership of Plos One, and would be suitable for publication after some minor revisions.

Thank you for your appreciation.

Introduction: The last paragraph in the introduction should give the reader more insight into your study. Based on previous studies, what are your hypotheses/expectations for each thing you examine?

We have modified the introduction accordingly. The new paragraph underlines our hypotheses and major findings more clearly. The modifications are coloured in the annotated version of the manuscript.

Discussion: The authors need to improve the discussion section, focusing on this study’s importance and relevancy.

We have added to the discussion.

Conclusion: Add suggestions and the actual necessities for future study.

We have added to the conclusion.

Reviewer 2

Proliferation: Only markers have been used to identify germ cell progenitors. Whether any sub-cultures could be done is not mentioned. No experiments to measure actual proliferation (like BrdU assay) is done. BrdU staining or assay should be done during different days of incubation to show that cells are actually proliferating in vitro. Mere presence of germ line markers cannot verify if actual proliferation is happening or not. So, according to me, manuscript does not do justice to the title.

Thank you for these comments. Although we did not use BrdU assays, the same wells were counted at different time points in Figure 1 and Figure 2. Since the total number of cells increases, it seems reasonable to assume that there was proliferation. Since the number of cells obtained from male mantle cells was orders of magnitude higher than in the other cultures, we used DDX4 and cytokeratin as markers. Together, these two antibodies stained more than 80% of the cells obtained in cultures of male mantle cells and more than 70% of those obtained from female mantle cells. Based on cell morphology, mainly the observation of a rosette-like array of five mitochondria at the base of most cells in male mantle cultures, we suspected that the cells surviving in culture were indeed of the germ cell lineage. DDX4 has been used in a large number of species, including mollusks, to identify cells from the germ cell lineages. It should also be reinforced that in mytilus edulis, gonadal tissue is intertwined in the mantle tissue. Since over time, in most assays, over 80%? of the cells were DDX4 positive, we did not try to identify other cell types in these cultures.

Gill cells indeed seemed to be more diverse based on morphology and contained a larger amount of cells that were not stained by either DDX4 or cytokeratin. However, since the number of cells and proliferation obtained from these samples was limited, testing a larger marker array was not possible.

Subcultures could not be achieved, as we could not detach the cells from the substrate with trypsin treatment. We did not pursue this line of investigation further and thus did not comment on the original submission. 

Inconsistency in methodology and results- Only 2 temperature mentioned in methodology, but results talk about 4 temperatures tested, but then goes on to elaborate only about two specific temperatures 15 and 25 oC. Methodology does not describe the three cell dissociation methods tested, but only the final chosen method is described.

We apologize for these inconsistencies. We wished to shorten the materials and methods sections by presenting only the protocols we selected and for which we provide results. We mentioned our earlier unsuccessful assays in the Results section with the aim of preventing others from repeating our mistakes. However, based on these comments, we now provided a thorough account of all the assays that we did, even if they did not yield any viable cell cultures.

(a) In the figure 1B and C, significant differences between days is not mentioned. A graph with mean and standard deviation, and significance values denoted as letters will be easier to interpret. The current graphs do not show any symbols to indicate if there were significant differences, and the graphs show overlapping values between the different days of observation (see figure 1C), creating doubt if there was actually any significant difference (i.e. if there was actually any significant cell proliferation). The authors say that cell proliferation (male mantle) occurred only until Day 15 like has been previously reported by Attaallah et al. 2020. In that case, what new findings do authors have to contribute?

Thanks for pointing that out. We have not added significance levels to figures 1B and 1C because the large number of groups resulting from the combination of different media and different incubation times would result in an unreadable graph. However, the statistical analysis is unambiguously showing significant differences between media and time. We provide the glm model as a supplementary figure. 

To better show whether the cells are proliferating, we made pair-wise comparisons between successive days that we provide as supplementary material as well. As we did not observe any cell growth at 15°C, it seemed useless to do this analysis for these data points. 

We thank the reviewer for the remark regarding our statement that proliferation occurred until Day 15 as in Attaalah et al. (2020). We have removed it. This was meant to underline a similitude between our results and those obtained by another group for another species but from similar tissues. However, there were many other differences, and they are best treated in the Discussion.

(b) Fig 2 A and B: In the study based on cell proliferation rate in different sex, the results say that female mantle cells also proliferated till Day 15. So, what is the difference in cell proliferation rate between male and female mantle cells? On what terms do you say that male mantle cells are better? Also, significant differences are not plotted on the graph. Also, many values in female mantle cells overlap with that of male mantle cells. The graph will be clearer with mean and standard deviation plots. You can keep either figure 2A or figure 2B. Two graphs convey more or less the same matter.

We apologize for not providing a better description of our results. Female mantle cells indeed showed proliferation; both male and female mantle cells showed greater proliferation than gill cells. However, relatively few female mantle cells are attached to the substrate compared to male cells, even with an increasing amount of donor tissue. We agree with the reviewer that Fig2A and 2B might be redundant. We have replaced both figures with a new Figure 2 where the number of cells is reported on a log10 axis, which allows a better comparison between samples. As for Figure 1, we provide another representation of the data denoting statistical differences between adjacent days as supplementary material.

(c) Immunofluorescence: Fluorescence measured on 1, 5 and 25. It is said that proportion of germ cells increased on Day 25, whereas cell counting experiment says cell proliferation occurred only until Day 15. These may be confounding results. It may not be termed proliferation, but just a shift in cell population. So, this statement is inappropriate “DDX4-positive cells were the only cells that significantly proliferated in culture, and only from male tissue samples”. Keep either Figure 5 or 6. Two graphs convey more or less the same matter. Again, letters to denote significant differences among groups are required.

The reviewer rightly points out that we cannot ascertain whether DDX–4 positive cells are proliferating or if cells that proliferated earlier are expressing DDX–4 by the time we measured its expression. We have rephrased the description of the results to reflect our findings more accurately. We have also removed Figure 5 and renumbered the figures accordingly.

Wide literature including personal experience shows that Amphotericin B is toxic to cells in vitro. How could you manage a long-term cell culture with Amphotericin B?

We are aware of these reports. Because contamination was an issue, we could not avoid using Amphotericin B. The concentration we used was tolerable for cells in other studies on bivalve mollusks. For example, Attaalah et al. used amphotericin B at 2.5 μg/mL. We tested several concentrations and found that cells could sustain 1.5 μg/mL.

Figure 3 is difficult to interpret using the arrowhead legends. Example, Type A cells are supposed to exhibit both antigens, but I cannot see both antigens in the representative picture they have shown in Figure 3 (Panel A). May be both antigens are present on the cells, but the picture is not clear. Similarly for other images.

The channels’ superposition might make the different labelling difficult to interpret. We now provide separate images for the different channels. Since this figure is now very large, we removed panels D-F.

The same modifications were made to Figure 4.

Figure 7 is a beautiful depiction of cell differentiation in vitro

Thank you for this comment.

Overall comment: the effort is worth appreciating but the results seem to be confounding, and the title and conclusions are misleading.

We thank the reviewer for his appreciation and hope that our revisions helped to clarify these issues,

---

## [Decision Letter · Decision Letter 1]

9 Jan 2024

In vitro proliferation of Mytilus edulis male germ cell progenitors

PONE-D-23-29958R1

Dear Dr. Angers,

We’re pleased to inform you that your manuscript has been judged scientifically suitable for publication and will be formally accepted for publication once it meets all outstanding technical requirements.

Kind regards,

Gao-Feng Qiu

Academic Editor

PLOS ONE

Additional Editor Comments (optional):

Reviewers' comments:

Reviewer's Responses to Questions

**Comments to the Author**

1. If the authors have adequately addressed your comments raised in a previous round of review and you feel that this manuscript is now acceptable for publication, you may indicate that here to bypass the “Comments to the Author” section, enter your conflict of interest statement in the “Confidential to Editor” section, and submit your "Accept" recommendation.

Reviewer #2: All comments have been addressed

2. Is the manuscript technically sound, and do the data support the conclusions?

Reviewer #2: Yes

3. Has the statistical analysis been performed appropriately and rigorously? 

Reviewer #2: Yes

4. Have the authors made all data underlying the findings in their manuscript fully available?

Reviewer #2: Yes

5. Is the manuscript presented in an intelligible fashion and written in standard English?

Reviewer #2: Yes

6. Review Comments to the Author

Reviewer #2: This manuscript on "In vitro proliferation of Mytilus edulis male germ cell progenitors" has been comprehensively improved. All comments from the reviewers were carefully considered and addressed point by point in the revised manuscript and I have no further comments. Also, thaks to the correspomding author to give better clarification on my queries; as i too in the similar area to develop a cell lines from invertebrates, whcich is really a much difficult area in cell culture research. Please carefully check the format of the entire article, for example, the scientific name needs to be italic (mainly in the reference section).

7. PLOS authors have the option to publish the peer review history of their article (what does this mean?). If published, this will include your full peer review and any attached files.

Reviewer #2: **Yes: **Dr. Jayesh Puthumana

---

## [Editor Report · Acceptance letter]

30 Jan 2024

PONE-D-23-29958R1 

PLOS ONE

Dear Dr. Angers, 

I'm pleased to inform you that your manuscript has been deemed suitable for publication in PLOS ONE. Congratulations! Your manuscript is now being handed over to our production team.

Kind regards, 

on behalf of

Prof. Gao-Feng Qiu 

Academic Editor

PLOS ONE